# Essential Oil from *Eucalyptus globulus* (Labill.) Activates Complement Receptor-Mediated Phagocytosis and Stimulates Podosome Formation in Human Monocyte-Derived Macrophages

**DOI:** 10.3390/molecules27113488

**Published:** 2022-05-28

**Authors:** Manuela Zonfrillo, Federica Andreola, Ewa K. Krasnowska, Gianluca Sferrazza, Pasquale Pierimarchi, Annalucia Serafino

**Affiliations:** Institute of Translational Pharmacology, National Research Council of Italy, 00133 Rome, Italy; manuela.zonfrillo@ift.cnr.it (M.Z.); federica.andreola@ift.cnr.it (F.A.); ewa.krasnowska@ift.cnr.it (E.K.K.); gianluca.sferrazza@ift.cnr.it (G.S.); pasquale.pierimarchi@ift.cnr.it (P.P.)

**Keywords:** eucalyptus essential oil, complement receptor-mediated phagocytosis, podosomes, macrophages

## Abstract

*Eucalyptus* essential oil and its major constituent eucalyptol are extensively employed in the cosmetic, food, and pharmaceutical industries and their clinical use has recently expanded worldwide as an adjuvant in the treatment of infective and inflammatory diseases. We previously demonstrated that essential oil from *Eucalyptus globulus* (Labill.) (EO) stimulates in vitro the phagocytic activity of human monocyte-derived macrophages and counteracts the myelotoxicity induced by the chemotherapeutic 5-fluorouracil in immunocompetent rats. Here we characterize some mechanistic aspects underlying the immunostimulatory ability exerted by EO on macrophages. The internalization of fluorescent beads, fluorescent zymosan BioParticles, or apoptotic cancer cells was evaluated by confocal microscopy. Pro-inflammatory cytokine and chemokine release was determined by flow cytometry using the BD cytometric bead array. Receptor involvement in EO-stimulated phagocytosis was assessed using complement- or IgG-opsonized zymosan particles. The localization and expression of podosome components was analyzed by confocal microscopy and western blot. The main results demonstrated that: EO-induced activation of a macrophage is ascribable to its major component eucalyptol, as recently demonstrated for other cells of innate immunity; EO implements pathogen internalization and clearance by stimulating the complement receptor-mediated phagocytosis; EO stimulates podosome formation and increases the expression of podosome components. These results confirm that EO extract is a potent activator of innate cell-mediated immunity and thereby increase the scientific evidence supporting an additional property of this plant extract besides the known antiseptic and anti-inflammatory properties.

## 1. Introduction

Monocytic/granulocytic cells, as well as differentiated macrophages, represent the primary cellular effectors of the immune response, which have a crucial role in the recognition of foreign bodies such as viruses, bacteria, and fungal pathogens, that ultimately results in the phagocytosis and digestion of such pathogenic microorganisms [1]. Phagocytic cells can also ingest and eliminate apoptotic cells through a process termed “efferocytosis” [2], thus contributing to cell clearance and turnover in tissue homeostasis. Phagocytosis is coupled to intracellular signals that prompt cellular responses including alterations in membrane trafficking, cytoskeletal rearrangement, and the release of chemical mediators such as pro- and anti-inflammatory cytokines and chemokines [1]. The recognition of foreign particles and pathogens by phagocytes, particularly by macrophages, is achieved through different nonopsonic or opsonic receptors [3,4]. Specifically, to facilitate the phagocytosis of opsonized and nonopsonized particles, differentiated macrophages express the Fc receptors (FcR) that bind to the Fc portion of immunoglobulin; the complement receptors (CR), such as CR3 (or αM/β2 integrins, also known as CD11b/CD18) that bind to iC3b present on foreign particles after complement activation; the mannose receptor (MR), which recognizes mannose and fucose on the pathogen surface; and several kinds of scavenger receptors [1,3]. Pathogens are mainly internalized by CRs after relatively nonspecific opsonization with complement proteins or by FcR after specific opsonization with antibodies. There are important differences in the mechanisms underlying the phagocytosis mediated by these two different receptors that are currently recognized, including differences in the cytoskeletal components involved in internalization and in inflammatory responses. In particular, FcR-mediated phagocytosis is tightly coupled to the production and secretion of proinflammatory cytokines and chemokines, while CR-mediated phagocytosis may not be. Furthermore, only the CR-mediated internalization requires the integrity of the microtubular network [5].

Macrophage adhesion and migration are the other two functions, together with phagocytosis, that are essential for the detection and elimination of microorganisms, debris, and foreign bodies. Both adhesion and migration involve the formation and turn-over of highly dynamic, actin-rich adhesion structures called podosomes [6,7], for which the impairment of function can lead to reduced innate immunity and is associated with several immunodeficiency disorders [8,9]. Podosomes consist of dot-shaped contacts with two structural domains; an actin-rich core, also containing some intermediate filaments and actin regulators; and a surrounding ring region that contains actin-binding and actin-adhesive proteins, including integrins and integrin regulatory proteins [6].

The use of naturally derived aromatic oils, widely spread in modern cosmetics as well as in folk medicine, has recently been extended to the clinical setting for the treatment of various diseases, and particularly for different types of inflammatory diseases such as rheumatism, allergies, and arthritis. In particular, the essential oil from *Eucalyptus* plants (Myrtaceae) and its major component 1,8-cineole (also known as eucalyptol, constituting 60–90% of the essential oil depending on the species [10,11]) is extensively employed in the cosmetic, food, and pharmaceutical industries, and its clinical use has expanded worldwide as an adjuvant in infective and inflammatory diseases [12,13,14,15,16,17]. Aside from eucalyptol, other components of EO include some terpenes such α-pinene and limonene, present in percentages around 9% and 2%, respectively, and more than 40 other minor compounds present in concentrations lower than 0.2% or in traces [18]. It has been recently reported that the topical application of *Eucalyptus* oil suppresses edema and the enhancement of vascular permeability in IgE-mediated allergic dermatitis [19], and that the rational use of this essential oil can improve the immune function of the respiratory tract as well as the whole immune response [20,21]. *Eucalyptus* extracts have been also been evaluated among 39 herbal medicines to be used as adjuvants in symptomatic therapy in the context of COVID-19, and the benefit/risk assessment for the *Eucalyptus* essential oil was found promising, with an overall benefit classified as mild [22]. We previously demonstrated that the essential oil from *Eucalyptus globulus* Labillardière (EO) stimulates in vitro the phagocytic activity of human monocyte-derived macrophages (MDMs) and counteracts in vivo the myelotoxicity induced by the chemotherapeutic 5-fluorouracil in immune-competent rats [23]. EO-stimulated internalization is coupled with a low release of pro-inflammatory cytokines and requires the integrity of the microtubule network, suggesting that EO could act by stimulating the complement receptor-mediated phagocytosis [23]. These data demonstrated that *Eucalyptus* essential oil can implement the innate cell-mediated immune response and can provide scientific support for an additional application of this plant extract, aside from the already known antiseptic and anti-inflammatory properties.

In this work, we aimed to better characterize the previously demonstrated immunostimulatory ability exhibited by EO on macrophages [23] by exploring a set of effects exerted by this essential oil on different functions related to the macrophage-mediated immune response. Specifically: (i) we assessed whether EO-stimulated phagocytosis in macrophages is ascribable to its major component eucalyptol, as demonstrated for other cells of the immune system [20]; (ii) we evaluated whether the EO-stimulated internalization of microorganisms is coupled with an improvement of their digestion; (iii) we assessed whether EO can also improve efferocytosis; (iv) we explored the involvement of complement and/or IgG receptors in EO-stimulated phagocytosis; (v) we verified whether EO stimulation could affect podosome formation and macrophage motility.

## 2. Results

### 2.1. Comparative Efficacy of Eucalyptus Oil and Eucalyptol on Phagocytic Ability and Viability of MDMs

To assess if the EO-stimulated phagocytosis on MDMs could be ascribable to its major component eucalyptol, a comparative analysis of the effects exerted by EO vs. eucalyptol on macrophage phagocytic ability has been performed after the administration of fluorescent beads. As detailed in Materials and Methods, to obtain comparable doses of treatment between the whole extract and eucalyptol, the latter was used at concentrations equivalent to 60% and 80% (equal to 0.0048% and 0.0064% of eucalyptol, respectively) of the content of the dose used for the whole extract (0.008%).

The results reported in Figure 1A,B confirmed that EO (used as a comparative control) is able to significantly increase the phagocytic activity of MDMs vs. the untreated control, in terms of both the percentage of phagocytic cells and number of phagocyted beads/cell (*p* < 0.01), as previously published [23]. A qualitative similar effect vs. the untreated control has been recorded for both concentrations of eucalyptol used, even if eucalyptol yielded less efficient results when compared to the whole extract (Figure 1A,B). This might be due to the higher dose-dependent toxicity recorded for both concentrations of eucalyptol vs. EO (Figure 1D), that could negatively affect macrophage phagocytic ability. Nonetheless, the results obtained strongly suggest that the EO-stimulated phagocytosis is ascribable to eucalyptol, which, although more toxic when used as a single compound, exerts a similar stimulatory effect on macrophage phagocytic ability. Moreover, similar to what was recorded for EO and previously published [23], the eucalyptol-stimulated phagocytosis occurs (at least at 24 h of treatment) with a very low release of pro-inflammatory cytokines and chemokines in the extracellular medium (Figure 2).

This effect is particularly evident for those pro-inflammatory cytokines (IL-8, IL-6, and TNF-α) and chemokines (CXCL10/IP10, CCL2/MCP-1, and CCL5/RANTES) that are conspicuously produced by MDMs under LPS stimulation, used as a positive control of macrophage activation. However, some quantitative differences have been recorded for the CCL2/MCP-1 inflammatory chemokine released by EO or by eucalyptol-treated macrophages (Figure 2B). Specifically, even if a lower release of this chemokine was recorded for both treatments compared with the untreated control, MDMs treated with 0.0048% eucalyptol released a significantly (*p* < 0.05) higher amount of CCL2/MCP-1 than that recorded under the EO treatment (Figure 2B).

### 2.2. Effect of Eucalyptus Oil on Microorganism Clearance

To better characterize the immunostimulatory efficacy exerted by EO on MDMs, the ability of essential oil to improve not only the internalization but also the elimination of ingested microorganisms has been assessed. Towards this purpose, the macrophage cultures were pre-treated for 24 h with EO, subjected to challenging with fluorescently labeled nonopsonized zymosan (*S. cerevisiae*) BioParticles, and analyzed by confocal microscopy at 2.5 h, 6 h, and 24 h after particle administration, in comparison with the untreated control (Figure 3A). The quantitative evaluation of the percentage of phagocytes (Figure 3B) and the number of internalized zymosan particles/cell (Figure 3C), during the time course performed, showed that EO pre-treatment improves both the internalization and the clearance of yeast particles. Indeed, already 2.5 h after the addition of zymosan to the cultures, the number of phagocyted particles/cell in EO treated MDMs was 2.2-fold higher than the control at the same time point (*p* < 0.001; Figure 3C), and at 6 h and 24 h post-challenge, a significant increment in the percentage of phagocyting cells was also recorded, with an increment up to 1.5-fold in the EO-treated cultures vs. the untreated control (Figure 3B). More interestingly, when the medium was replaced at 6 h post-challenge with a fresh zymosan-free medium and the cells were analyzed after an additional 18 h, the number of internalized yeasts per cell in EO treated cultures was reduced by more than 40% compared with the untreated control at the same time point (Figure 3A, right panels; Figure 3C, red bars). This strongly suggests that in EO-treated MDMs the clearance of ingested microorganisms occurs faster than in untreated macrophages.

### 2.3. Effect of Eucalyptus Oil on Efferocytosis

The ability of essential oil to improve the MDMs’ phagocytic ability not only against non-specific foreign bodies (fluorescent beads) or microorganisms (yeasts or bacteria) but also towards apoptotic cells has also been assessed. To this purpose, apoptotic bodies (AB) obtained from the MCF7 human breast cancer cells by staurosporine treatment, as described in Materials and Methods, were stained with propidium iodide (PI), administered to EO pre-treated MDMs and untreated controls and analyzed after 1 h and 3 h (Figure 4). MDMs pre-treated for 6 h with 0.1 µg/mL LPS were used as a positive control of macrophage activation. Confocal microscopic observation (Figure 4A) and quantitative evaluation of the percentage of phagocytes (Figure 4B) as well as of the number of internalized AB/cells (Figure 4C) at the defined times showed that EO pre-treatment can stimulate the internalization of AB. Indeed, already 1 h after AB administration, the percentage of phagocyting macrophages under EO treatment was 2.2-fold higher compared to the untreated control at the same time, and this increase was maintained at 3 h when both the percentages of phagocytes and the number of AB/cells were also significantly higher than those recorded in LPS-treated MDMs (Figure 4B,C).

### 2.4. Assessment of the Involvement of Complement Receptor in EO-Stimulated Phagocytosis

To determine if EO promotes complement receptor (CR)- or IgG receptor (FcR)-mediated phagocytosis, a phagocytosis assay was performed that uses fluorescent complement-opsonized (COZ) and IgG-opsonized (IgOZ) zymosan particles, respectively. For this assay, Alexa Fluor 488-IgOZ (green hue) and Alexa Fluor 594-COZ (red hue) were simultaneously added at a ratio of 20 yeast/cell and with a COZ/IgOZ ratio of 1:1 to the MDM cultures pre-treated 24 h with 0.008% EO or 6 h with 0.1 μg/mL LPS, the latter used as a positive control of macrophage activation. Samples were analyzed 1.5 h and 3 h after the addition of zymosan particles to the cultures. As shown in Figure 5, in the untreated control, IgOZ particles were preferentially internalized at both times analyzed, with a ratio COZ(n)/IgOZ(n) ≅ 0.5 (Figure 5D). 1 h post-challenge, LPS treatment promoted the internalization of both COZ and IgOZ particles [ratio of COZ(n)/IgOZ(n) ≅ 1], while COZ particles were preferentially phagocytosed 3 h post-challenge [ratio COZ(n)/IgOZ(n) ≅ 1.7]. Conversely, in EO-treated cultures, COZ particles were internalized at a number which was 2.1- and 3.3-fold higher than the IgOZ particles after 1.5 h and 3 h, respectively (Figure 5D), indicating that the essential oil mainly acts by promoting the CR-mediated phagocytosis. Furthermore, 3 h post-challenge, the number of COZ particles/phagocytes was significantly (*p* = 0.001) higher in EO-treated MDMs compared with LPS-treated cultures analyzed at the same time (Figure 5C).

### 2.5. Effect of Eucalyptus Oil on Podosome Formation and Podosome Structural Components

To assess whether EO affects podosome formation, the association of the actin-binding protein vinculin with the actin-rich dot-shaped contacts at the cell adhesion sites was firstly analyzed. Confocal microscopic analysis showed that in EO-treated MDMs actin/vinculin double-stained podosome structures were highly increased compared with the untreated control (Figure 6A). This was confirmed by the quantitative analysis that recorded a significant increment in podosome density per cell in the EO-treated vs. the untreated cultures (2.5-fold vs. control; *p* < 0.05) (Figure 6C). Further, in macrophages treated with the essential oil, the colocalization index (CI) of the actin cores and vinculin-rich rings (yellow hue in Figure 6 A and white hue in Figure 6B) increased 1.6-fold compared to the untreated control (Figure 6D; *p* < 0.05).

The ability of EO to affect the expression of αM (CD11b) and β2 (CD18) integrins, two other main structural components of the podosome ring that alongside other integrins regulate the interaction between the cell and the extracellular matrix [6], has been also verified. WB analysis performed in a time-course experiment from 1 h to 24 h (Figure 7) showed that EO-stimulated MDMs expressed up to about twofold higher levels of αM integrin (1.8-fold vs. the untreated control at 24 h, *p* < 0.01) compared to the untreated control, with a significant increase recorded already 1 h after EO addition to the cell culture medium. Moreover, the expression levels of β2 (CD18) integrin significantly increased after 3 h and 6 h of EO treatment (1.62-fold and 1.44-fold vs. control, respectively; *p* < 0.05), time points at which both αM and β2 integrins were concomitantly augmented (Figure 7).

### 2.6. Effect of Eucalyptus Oil on Macrophage Motility

To assess whether EO affects the migratory ability of macrophages in response to a chemotactic stimulus, a Transwell migration assay, in the absence or in the presence of 0.008% EO, was performed as described in Materials and Methods. MCSF-1 (40 ng/mL) was used as a chemoattractant. As reported in Figure 8, after 48 h from seeding, the number of EO-stimulated cells recovered at the bottom of the inserts was significantly (*p* < 0.001) higher compared to the untreated control (2.3-fold vs. control), indicating that EO treatment had increased macrophage migration through the porous membrane insert. No significant (*p* = 0.1556) differences were recorded after 24 h from seeding (Figure 8).

## 3. Discussion

*Eucalyptus* essential oil was originally used by Australian aboriginals as a remedy for inflammation and wounds [24], and for its anti-inflammatory, analgesic, sedative, and bactericidal effects, and it is currently used in aromatherapy, herbal medicine, and in the clinical practice as an adjuvant for the treatment of infective and respiratory diseases [17,25,26,27]. In the last decade, various independent studies reported that EO, as well as its main constituent eucalyptol, might be beneficial for the cell-mediated immune response against infectious diseases, immunosuppressive pathologies, or after anticancer therapy [20,21,23,28,29], but the mechanism underlying this immunostimulatory activity has been poorly investigated.

Starting from our previously published results [23], in this work, we better characterize some of the mechanistic aspects underlying the ability of EO to improve the functions of cells implicated in innate immunity, and in particular macrophages. Firstly, in a comparative study of the efficacy of EO vs. eucalyptol treatments, we showed that eucalyptol, used at concentrations comparable to those contained in the whole extract, stimulates phagocytosis by MDMs similarly to EO, even if it exhibits a higher toxicity, suggesting that in the whole extract some minor component(s) that are able to preserve macrophage viability could be present. The higher cytotoxicity of eucalyptol at the doses used (0.0048% or 480 μM, and 0.0064% or 640 μM) is not surprising, since it has been recently reported that concentrations of eucalyptol higher than 50 µM were found toxic for platelets [30]. Furthermore, similar to EO, the eucalyptol-stimulated phagocytosis is coupled with a low release of pro-inflammatory cytokines and chemokines, and this is in line with the anti-inflammatory properties already reported of both EO and its main component. These results indicate that the EO-induced activation of macrophages previously demonstrated [23] is ascribable to its major component eucalyptol, as recently demonstrated for other cells of innate immunity [20].

The elimination of pathogens through phagocytosis operated by the cells of the innate immune system is a complex process that implies not only pathogen recognition and internalization into phagocytes but also their digestion. On this basis, our results demonstrating that EO might also improve the digestion by macrophages of phagocytized zymosan particles, and, more in general, of other microbial agents, reinforce the attractiveness of this essential oil as an immunostimulatory extract potentially useful as an adjuvant for the treatment of infectious and immunosuppressive diseases, even if further insights are needed to exhaustively clarify the molecular mechanism underlying such an improvement in killing potential.

The reported data, demonstrating that EO stimulates MDM phagocytic capacity towards apoptotic bodies, suggest that this essential oil could also be helpful for improving efferocytosis in both physiological and pathological conditions, and extend the validity of EO as possessing possible therapeutic benefits for the restoration and maintenance of tissue homeostasis, as an adjuvant in the treatment of diseases related to defective efferocytosis, or for the elimination of dying tumor cells after chemotherapy.

As it concerns the mechanism(s) through which EO stimulates macrophages, the phagocytosis assay, performed using the complement- and IgG-opsonized particles, definitively demonstrates that *Eucalyptus* oil implements pathogen internalization mainly by stimulating CR-mediated phagocytosis very shortly after contact with foreign microorganisms. This further reinforces the efficacy of this essential oil in stimulating the innate immune system during the early stages of pathogen infection.

We also obtained evidence indicating that EO might affect macrophage motility and chemotaxis by stimulating the assembly and disassembly of podosome structures. Indeed, EO stimulation significantly increases podosome density, as well as augmenting colocalization at the podosomes of the actin core with the ring component vinculin, which is known to function, together with talin and paxillin, as a linker between integrins and actin-associated proteins during podosome assembly [6]. The augmented colocalization recorded suggests that EO could improve the actin-binding capacity of vinculin [31], but further study is required to deeply clarify this point. Furthermore, at early stages of treatment EO induces a moderate but significant increase in the expression levels of αM integrin (CD11b) and β2 (CD18) integrins. Since these integrins are not only two of the main structural components of the podosome ring that regulate the interaction between cell and extracellular matrix during cell movement [26] but also form together the heterodimeric complex CD11b/CD18 (or CR3), this result provides further evidence that EO might stimulate podosome formation, macrophage motility, and the CR-mediated macrophagic functions. The Transwell migration assay confirmed that the essential oil can implement macrophage motility under a chemotactic stimulus. These results are stimulating for future insights aimed to assess whether this essential oil could also be proposed as a possible adjuvant for those immunodeficiency disorders in which a reduced capacity or an inability to make podosomes severely prejudices the migration and chemotaxis of macrophages and monocytes [8,9].

## 4. Materials and Methods

### 4.1. Preparation of the Human MDM Cultures

Peripheral blood mononuclear cells (PBMCs) were isolated from buffy coats from anonymized healthy blood donors, provided by the Transfusion Center of Policlinico “Tor Vergata” in Rome by density gradient centrifugation using the Lympholyte-H (Cedarlane, Hornby, ON, Canada) and stored, soon after isolation, in a liquid nitrogen cell bank. At the time of blood donation, all subjects gave their informed consent according to national and international guidelines. For the experiments, the lymphocytic/monocytic fraction was resuscitated and resuspended in an RPMI 1640 medium (Hyclone Labs Inc. Logan, UT, USA) supplemented with 20% (*v*/*v*) heat-inactivated fetal bovine serum (FBS; Hyclone Labs Inc. Logan, UT, USA), L-glutamine (2 mM), penicillin (100 IU/mL), and streptomycin (100 mg/mL). Cells were seeded on 175 cm^2^ flasks and maintained at 37 °C in 5% CO_2_. After 1 h of culture, non-adhering cells were removed and the residual adhering MDMs were maintained in culture for 7 days to obtain partially differentiated macrophages. Cells were then detached using cold PBS, seeded at a density of 2.5 × 10^4^ cells/cm^2^ in 35 mm culture plates or on cover-slips for the microscopic analyses, and allowed to adhere for 4–5 days before treatments. As previously published [32], after 10-12 days of adhesion, MDM cultures resulted almost exclusively represented by the CD45+ and CD14+ monocytic fraction and expressed high levels of CD44 on the cell surface, which are all features of mature macrophages.

### 4.2. MDM Treatments and Cell Viability Determination

Essential oil from *Eucalyptus globulus* was commercial and purchased from Sigma-Aldrich (St. Louis, Mo, USA; product number W246603, CAS number 84625-32-1). The percentage of 1,8-cineole was ≥70%, as reported in the product specification sheet. To exclude the presence of any endotoxins in the essential oil preparation used, the EO extract was preliminarily tested using the Limulus Amebocyte Lysate (LAL) test (PYROGENT. Plus—Lonza Walkersville, Inc., Walkersville, MD, USA), as previously reported [23]. MDMs, maintained in RPMI 1640 plus 10% (*v*/*v*) FBS, were treated for intervals ranging from 1 h to 24 h, with 0.008% *v/v* (corresponding to about 50 µg/mL) of EO or with 0.0048% and 0.0064% *v/v* (corresponding to 480 μM and 640 μM, respectively) of eucalyptol (1,8-cineole; Sigma-Aldrich Co., St. Louis, Mo, USA), concentrations equivalent to the 60% and 80% of the content of the dose used for the whole extract, respectively. These percentages of eucalyptol have been chosen assuming that 60% (0.0048%) and 80% (0.0064%) were equivalent to a lower and a higher dose, respectively, compared to the 1,8-cineole contained in the EO used (≥70%). The concentration of 0.008% was selected in preliminary dose-response experiments previously published [23]. MDMs stimulated with 0.1 µg/mL of bacterial lipopolysaccharide (LPS, Sigma-Aldrich Co., St. Louis, Mo, USA) for 6 h were used as a positive control for macrophage activation. Cell viability after EO and eucalyptol treatments was determined by the Trypan blue dye exclusion method.

### 4.3. Phagocytic Activity and Killing Ability Assays by Confocal Microscopy

The phagocytic activity of treated and untreated MDMs was tested by confocal microscopy, using the following methods:

(1) For the comparative analysis of the efficacy of EO vs. eucalyptol, by adding 2 × 10^7^ beads/mL of yellow-green fluorescent polystyrene beads to the cultures (∅ 1 μm, at a ratio of at least 10 beads/cell; Molecular Probes) as previously described [23]. Samples were analyzed after 30 min challenge.

(2) For the evaluation of the effect of EO on microorganism clearance, by evaluating, under a confocal microscope, the ingestion of nonopsonized fluorescent zymosan A (*S. cerevisiae*) BioParticles (Molecular Probes, Eugene, ONT, USA) covalently labeled with Alexa Fluor 488 (Ec/Em: 495/519 nm, green fluorescence). Zymosan particles were added to the cultures at a ratio of 20 particles/cell and analyzed at 2.5 h, 6 h, and 24 h post-challenge. For evaluating the killing ability of MDMs, in a parallel experiment, 6 h post-challenge the medium was replaced with a fresh zymosan-free medium, and cells were analyzed after additional 18 h.

For both assays, at the defined times, cells were fixed with 4% paraformaldehyde, counterstained with 1 µg/mL propidium iodide (PI—Sigma-Aldrich Co., St. Louis, MO, USA) and observed under a LEICA TCS SP5 confocal microscope (Leica Instruments, Heidelberg, Germany). A minimum of 200 cells per sample were evaluated, and the number of phagocytic MDMs (reported as a percentage of phagocytic cells), as well as the number of beads or zymosan per cell, were counted in a blinded fashion. The experiments were repeated three times and the mean values were plotted.

### 4.4. Generation of Apoptotic Bodies and Evaluation of Their Internalization by MDMs

Apoptotic bodies (AB) were obtained from the MCF7 human breast cancer cells via treatment with 4µM of the protein-kinase inhibitor staurosporine (Sigma Aldrich Co., St. Louis, MO, USA) for 18 h, with the dose and time of treatment selected in preliminary experiments. The phagocytic activity of MDMs pre-treated for 24 h with 0.008% EO and of untreated controls was tested by adding PI-stained apoptotic bodies to the cultures in a ratio of about 5 AB/cell. Samples were analyzed 1 h and 3 h after AB administration when cells were fixed with 4% paraformaldehyde and observed by confocal microscopy. Differential interference contrast (DIC) was used to visualize MDM morphology. A minimum of 200 cells per sample were evaluated, and the number of phagocytic MDMs and the number of AB per cell were counted in a blinded fashion. The experiments were performed in triplicate and the mean values were plotted.

### 4.5. Evaluation of Cytokine and Chemokine Release by MDMs

The concentrations of the human pro-inflammatory cytokines (IL-8, IL-6, TNF-α) and chemokines (CXCL10/IP10, CCL2/MCP-1, CXCL9/MIG, CCL5/RANTES) released into the culture media by EO and eucalyptol treated MDMs were determined using the BD cytometric bead array human inflammation flex set (BD Pharmingen, San Diego, CA, USA) according to the manufacturer’s protocol, as previously reported [23,32]. The threshold of the sensitivity of the assay ranged from 5 to 10 pg/mL for all cytokines and chemokines tested, as proved by the respective standard curves. MDMs stimulated with LPS were used as a positive control for macrophage activation. Flow-cytometric analysis was performed using a FACSCalibur flow cytometer (Becton Dickinson, Mountain View, CA, USA).

### 4.6. Phagocytosis Assay Using Complement- or IgG-Opsonized Zymosan BioParticles

Fluorescent zymosan particles were covalently labeled with Alexa Fluor 488 (Ex/Em: 495/519 nm, green fluorescence) or Alexa Fluor 594 (Ex/Em: 590/617 nm, red fluorescence), and opsonized with IgG (IgG-opsonized zymosan—IgOZ) or with complement (complement-opsonized zymosan—COZ), respectively. Internalization of MDMs of fluorescent zymosan particles was analyzed by confocal microscopy. Zymosan A BioParticles (*Saccharomyces cerevisiae*) were reconstituted and used as indicated by the manufacturer. For IgOZ particles, zymosan particles were opsonized using the opsonizing reagent (Molecular Probes) containing the rabbit polyclonal IgG antibodies specific for *S. cerevisiae*, according to the manufacturer’s instruction. For COZ particles, zymosan particles were opsonized with complement components by incubation at 37 °C for 1 h in fresh FBS, as previously described [5,32]. The opsonized particles were simultaneously administered (20 total yeasts/cell, ratio COZ/IgOZ 1:1) to EO or LPS pre-treated MDMs and untreated control, and samples were analyzed at 1.5 h and 3 h post-challenge. A minimum of 200 cells per sample were evaluated, and the number of phagocytic MDMs, as well as the number of phagocytized COZ and IgOZ per cell, were counted. The quantitative analysis was carried out in a blinded fashion.

### 4.7. Western Blot (WB) Analysis

MDM cultures were washed with ice-cold PBS and then lysed using 50 mM Tris-HCl buffer (150 mM NaCl, 1% NP-40, 10% glycerol, 0.1 mM EGTA, 0.5 mM EDTA, 50 mM NaF, 1 mM Na_3_OV_4_, and a protease inhibitor cocktail; Sigma-Aldrich) at pH 8.0. Lysates were clarified by centrifugation and protein content was determined using Bradford reagent (BioRad, Segrate, Italy). 15–20 µg of protein extract were separated using an 8% SDS/PAGE and then transferred to nitrocellulose membrane (Hybond, Amersham GE Healthcare, Chicago, Illinois, USA). Membranes were then blocked for 1 h at room temperature with 5% BSA in Tris-buffered saline-Tween (TBS-T; 0.02M Tris, 0.150M NaCl, pH 7.6 and 0.05% Tween-20), probed with the following antibodies: the rabbit monoclonal antibodies against integrin αM (CD11b/ITAM, Millipore Merck KGaA, Darmstadt, Germany; working dilution 1:2000), and the mouse monoclonal antibodies against β2 integrin (CD18, Enzo Life Sciences, NY, USA; working dilution 1:1000). Primary antibodies were revealed with peroxidase-conjugated secondary antibody (BioRad, Richmond, CA, USA; working dilution 1:3000). GAPDH, used as a loading control, was revealed using a rabbit polyclonal antibody from Santa Cruz Biotechnology (working dilution 1:20,000). Densitometric analysis was carried out using the ImageJ processing program (website for the ImageJ download: http://rsbweb.nih.gov/ij/ accessed on 28 February 2022). Uncalibrated Optical Density values, normalized to GAPDH, were reported as fold vs. untreated control. Results are as the mean of three independent experiments ± S.D.

### 4.8. Transwell Migration Assay

Cell migration assay was performed using the Falcon^®^ Permeable inserts with Transparent PET Membrane (8.0 µm pore size; Corning, product # 353097), following the manufacturer’s instructions. MDMs, detached and resuspended in medium containing 0.008% EO or in EO-free medium (untreated control), were seeded in the upper wells of the inserts and incubated at 37 °C, 5% CO_2_ atmosphere. Before incubation, 40 ng/mL of macrophage CSF 1 (MCSF-1) was added in the bottom wells to test chemotaxis. After 24 h and 48 h of incubation, the non-migrated cells were removed from the upper surface of the membrane by scrubbing and the residual cells that migrated to the bottom of the well were stained with the Giemsa stain solution (Bio-Optica Milano, Italy) and analyzed under the MOTIC AE31 optical microscope equipped with a digital camera. A quantitative evaluation of the number of migrated cells was performed on images obtained by exploring the bottoms of the wells, using the ImageJ processing software. The analysis was done on samples in triplicate and results were reported as mean values ± S.D.

### 4.9. Statistical Analysis

Statistical analyses were conducted using the two-tailed Student’s *t*-test or one-way analysis of variance (ANOVA) followed by Dunnett’s Comparison Test, to analyze significance vs. the untreated control, or by Tukey post-hoc test, to analyze the differences between groups. A *p* value < 0.05 was assumed as statistically significant. All data were obtained from at least three independent experiments performed on MDMs from three different healthy donors and presented as means ± SD. All statistical analyses were performed using PRISM Software, v5.04.

## 5. Conclusions

The majority of the knowledge on the therapeutic use of plant-derived essential oils was acquired through folklore, and only some properties of these natural extracts are actually supported by scientific studies. Since natural products are a valuable source for drug discovery, the exploration of the biological actions of plants’ medicinal extracts and the deepening of the mechanisms underlying these actions may drive the search for novel drugs. On this basis, our data, demonstrating that *Eucalyptus* essential oil from *Eucalyptus globulus* (as well as its main component eucalyptol) is able to stimulate the innate cell-mediated immune response, and specifically the macrophagic functions, provide further insight into our previously published data [23] and add scientific evidence supporting an additional property of this plant extract besides the known antiseptic and anti-inflammatory properties. Furthermore, the data demonstrate that EO can promote efferocytosis and extend the possible utility of this essential oil to age-related conditions involving the dysregulation of tissue homeostasis, and the data will thus stimulate further studies aimed to explicate this particular EO property. 

## Figures and Tables

**Figure 1 molecules-27-03488-f001:**
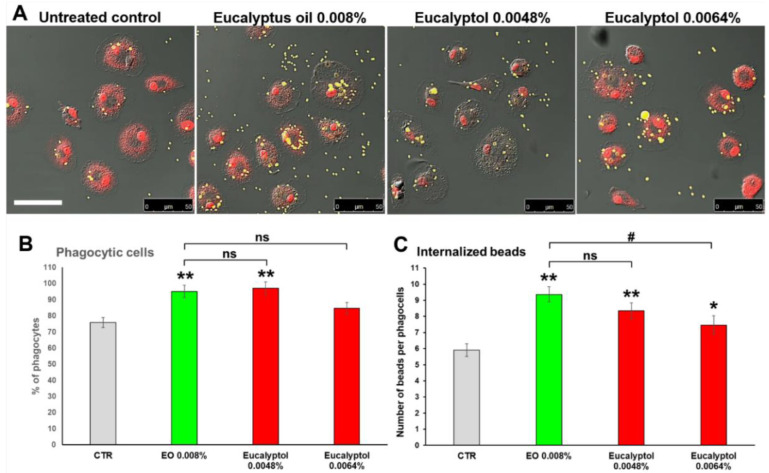
Comparative effects of *Eucalyptus* oil and 1,8-cineole (eucalyptol) treatments on phagocytic activity and viability of human MDMs. The phagocytic activity of treated and untreated MDMs was tested by adding to cultures 2 × 10^7^ beads/mL of fluorescent polystyrene beads as described in Materials and Methods. (**A**) Confocal microscopy images, showing the beads uptake (yellow hue) in untreated control and MDMs treated for 24 h with *Eucalyptus* oil or eucalyptol, the latter used at concentrations equivalent to 60% and 80% (equal to 0.0048% and 0.0064% of eucalyptol, respectively) of the content of the dose used for the whole extract (0.008%); cell nuclei were counterstained with propidium iodide (red hue). Merged images with differential interference contrast, used to visualize cell morphology, are shown. Bar = 50 µm. (**B**,**C**) Quantitative assessment of MDM phagocytic activity obtained by counting the number of phagocytes (**B**), reported as percentage of phagocytic cells, as well as the number of beads per cell (**C**). A minimum of 200 cells/sample were analyzed and the results are the mean ± SD from three independent experiments (*n* = 3). (**D**) Cell viability assay performed on MDMs cells after 24 h of EO and eucalyptol treatments by Trypan blue dye exclusion method; results, reported as a percentage of dead cells, are the mean ± SD from three independent experiments (*n* = 3). (**B**,**C**) Significance (One-way ANOVA + Tukey multiple comparison test), * vs. CTR: * *p* < 0.05, ** *p* < 0.01, *** *p* < 0.001; # vs. EO: # *p* < 0.05, ### *p* < 0.001; ns: not significant.

**Figure 2 molecules-27-03488-f002:**
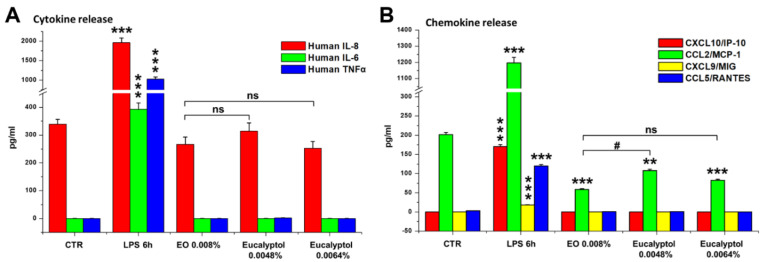
Effect of *Eucalyptus* oil and 1,8-cineole (eucalyptol) treatments on pro-inflammatory cytokines (**A**) and chemokines (**B**). Analysis by cytometric bead array of the human cytokines IL-8, IL-6, IL-10, and TNF-α (**A**), and chemokines CXCL10/IP10, CCL2/MCP-1, CXCL9/MIG, and CCL5/RANTES (**B**) released in culture medium by the untreated control (CTR); MDMs stimulated for 6 h with LPS and cells treated for 24 h with 0.008% EO, 0.0048% or 0.0064% eucalyptol. Significance (One-way ANOVA + Tukey multiple comparison test), * vs. CTR: ** *p* < 0.01, *** *p* < 0.001; # vs. EO: # *p* < 0.05; ns: not significant.

**Figure 3 molecules-27-03488-f003:**
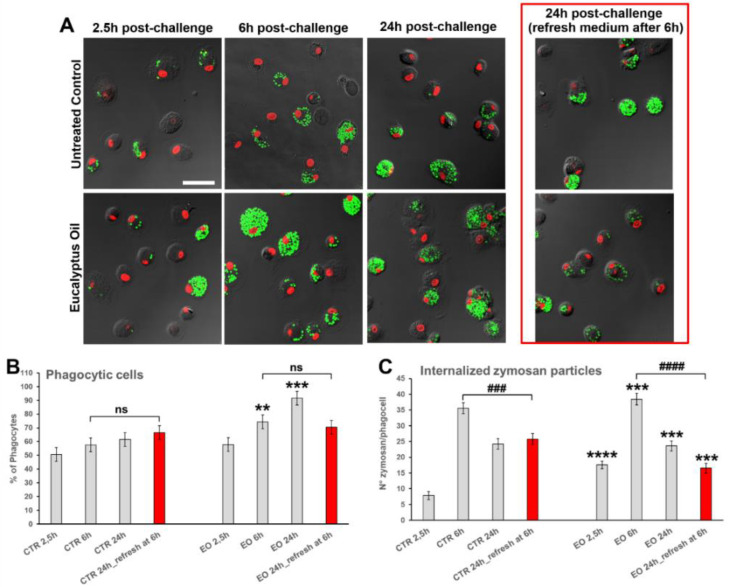
Effects of *Eucalyptus* oil treatment on phagocytic activity and clearance of zymosan (*S. cerevisiae*) particles by human MDMs. The phagocytic activity of MDMs pre-treated for 24 h with 0.008% EO and of untreated controls was tested by adding to cultures Alexa Fluor 488-labeled zymosan (green hue) in a ratio of 20 particles/cell, as described in Materials and Methods. Samples were analyzed 2.5 h, 6 h, and 24 h after zymosan particle administration. To evaluate the ability of MDMs to eliminate the yeast, in a parallel experiment, 6 h post-challenge the medium was replaced with fresh zymosan-free medium, and cells were analyzed after additional 18 h. (**A**) Confocal microscopy images, showing the zymosan uptake (left panels) and clearance (right panels in the red frame) in untreated control and EO treated MDMs. Cell nuclei were counterstained with propidium iodide (red hue). Merged images with differential interference contrast, used to visualize cell morphology, are shown. Bar = 50 µm. (**B**,**C**) Quantitative assessment of MDM phagocytic activity obtained by counting the number of phagocytes, reported as percent of phagocytic cells (**B**), as well as the number of zymosan particles per cell (**C**). The red bars in the graphs highlight the values recorded after removal of yeasts from culture media at 6 h post-challenge. A minimum of 200 cells/sample were analyzed and the results are the mean ± SD from three independent experiments (*n* = 3). Significance (two-tailed Student’s *t*-test), * vs. CTR at the same time: ** *p* < 0.01, *** *p* < 0.001, **** *p* < 0.0001; # vs. values at 6 h before removal of zymosan from culture media: ### *p* < 0.001, #### *p* < 0.0001; ns: not significant.

**Figure 4 molecules-27-03488-f004:**
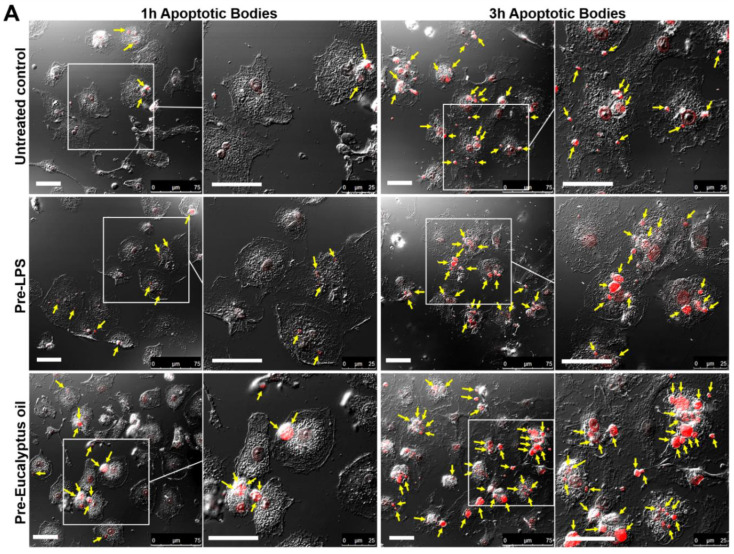
Effects of *Eucalyptus* oil treatment on MDM phagocytic ability against apoptotic bodies from the MCF7 human breast cancer cells. Apoptotic bodies (AB) were obtained from MCF7 breast carcinoma cells as described in Materials and Methods. The phagocytic activity of MDMs pre-treated for 24 h with 0.008% EO and of untreated controls was tested by adding to cell cultures the PI-stained apoptotic bodies (AB; red hue) in a ratio of about 5 AB/cell. MDMs pre-treated for 6 h with 0.1 µg/mL LPS were used as a positive control of macrophage activation. Samples were analyzed 1 h and 3 h after AB administration. (**A**) Confocal microscopy images, showing the AB uptake in untreated control and EO treated MDMs. Merged images with differential interference contrast, used to visualize cell morphology, are shown. Arrows point to internalized AB. Bar = 50 µm. (**B**,**C**) Quantitative assessment of MDM phagocytic activity obtained by counting the number of phagocytes, reported as percent of phagocytic cells (**B**), as well as the number of AB per cell (**C**). A minimum of 200 cells/sample were analyzed and the results are the mean ± SD from three independent experiments (*n* = 3). Significance (One-way ANOVA + Tukey multiple comparison test), * vs. CTR at the same time: ** *p* < 0.01, *** *p* < 0.001; # vs. LPS: # *p* < 0.05, ### *p* < 0.001; ns: not significant.

**Figure 5 molecules-27-03488-f005:**
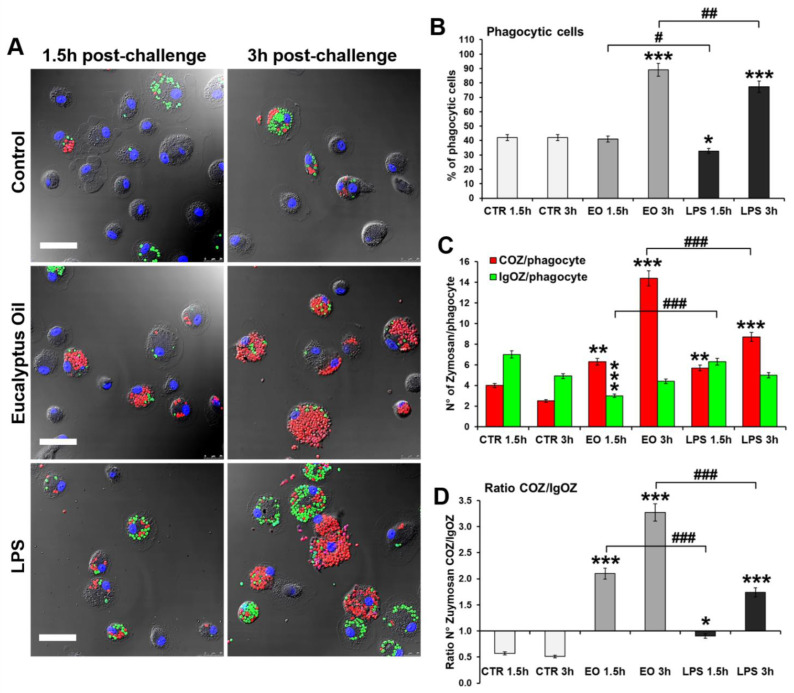
Complement receptor involvement in EO-stimulated phagocytosis. (**A**) Representative images by confocal microscopy of MDM cultures 1.5 h and 3 h after addition of fluorescent COZ or IgOZ to the culture medium. Alexa Fluor 594-COZ (red hue) and Alexa Fluor 488-IgOZ (green hue) were simultaneously added (20 total yeasts/cell, ratio COZ/IgOZ 1:1) to MDMs pre-treated 24 h with 0.008% EO or 6 h with 0.1 µg/mL LPS. Cell nuclei were counterstained with 0.2% Hoechst. Merged images with differential interference contrast, used to visualize cell morphology, are shown. Bars = 50 μm. (**B**–**D**) Quantitative analysis of the phagocytosis assay: results were obtained by counting the number of phagocytes, reported as percent of phagocytic cells (**B**) as well as the number of red COZ or green IgOZ per phagocytic cell (**C**), also expressed as ratio COZ (n)/IgOZ (n), for which values >1 indicate that COZ particles are internalized at higher numbers than the IgOZ particles. A minimum of 200 cells/sample were analyzed and the results are the mean ± SD from three independent experiments (*n* = 3). Significance (One-way ANOVA + Tukey multiple comparison test); * significance vs. CTR at the same time: * *p* < 0.05, ** *p* < 0.01, *** *p* < 0.001; # significance vs. LPS at the same time: # *p* < 0.05, ## *p* < 0.01, ### *p* < 0.001.

**Figure 6 molecules-27-03488-f006:**
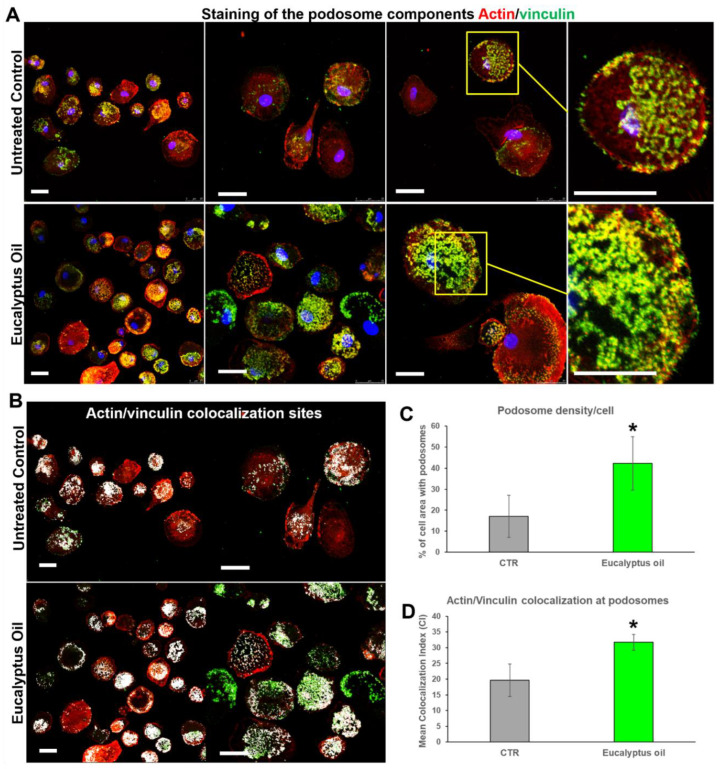
Effect of *Eucalyptus* oil on actin and vinculin distribution at the podosomes of human MDM. (**A**) Confocal microscopy showing the association of vinculin (green hue) with the actin-rich dot-shaped contacts (red hue) at the cell adhesion sites (yellow hue) of macrophages treated for 24 h with 0.008% EO and of untreated control. Cells were double-stained for F-actin, using the TRITC-phalloidin, and for vinculin, using the specific monoclonal antibody, as described in Materials and Methods. Cell nuclei were counterstained with 0.2% Hoechst. Fields at increasing magnification are shown. Bars = 25 μm. (**B**,**D**) Confocal microscopy (**B**) showing the colocalization regions (white areas) of actin and vinculin at the podosomes, and a bar graph (**D**) reporting the mean colocalization indexes (CI) calculated as described in Materials and Methods, in control and EO-stimulated MDMs. Bars = 25 μm. (**C**) Quantitative analysis of podosome density/cell (percent of total cell area occupied by podosomes) in control and EO-stimulated MDMs, performed by using the ImageJ processing software. Quantification of CI and the podosome density have been carried out by analyzing a minimum of 100 cells/sample and results are the mean ± SD from three independent experiments (*n* = 3). Significance (two-tailed Student’s *t*-test) vs. CTR, * *p* < 0.05.

**Figure 7 molecules-27-03488-f007:**
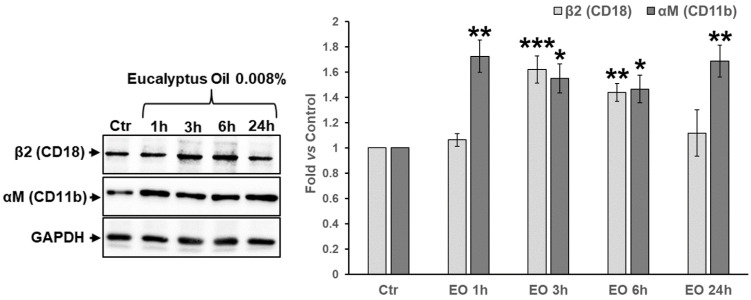
Time course of the expression of β2 (CD18) and αM (CD11b) integrins in EO-stimulated MDMs. WB analysis (left panel) of the expression levels of αM and β2 integrins after 1 h, 3 h, 6 h, and 24 h of treatment with 0.008% EO. Results from the densitometric analysis (right panel) of the expressions of the two podosomal components and CR subunits, normalized to GAPDH, are also reported. Results, reported as fold vs. untreated control, are the mean ± SD from three independent experiments performed on MDMs from three different donors (*n* = 3). Significance (One-way ANOVA + Dunnett’s Comparison Test), * vs. CTR; * *p* < 0.05, ** *p* < 0.01, *** *p* < 0.001.

**Figure 8 molecules-27-03488-f008:**
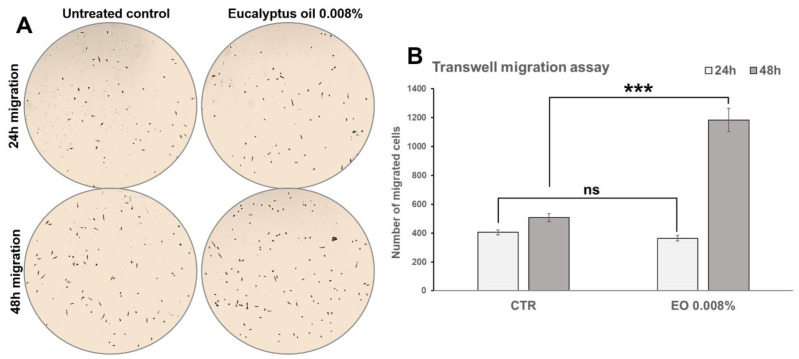
Effect of Eucalyptus oil on macrophage motility by Transwell migration assay. (**A**) Representative images of Giemsa-stained cells recovered at the bottom of the inserts after 24 h and 48 h of incubation in absence (untreated control) and in presence of 0.008% EO. (**B**) Bar graph reporting the quantification of number of cells recovered at the bottom of the inserts after 24 and 48 h from the seeding of untreated and in EO-stimulated MDMs. The assay was performed on samples in triplicate (*n* = 3), and results are reported as mean values ± S.D. Significance vs. CTR (two-tailed Student’s *t*-test): *** *p* < 0.001; ns: not significant.

## Data Availability

All the data produced in this study are reported in this article. The primary data files are available from the corresponding author upon reasonable request.

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
