# Peer review of "Essential Oil from Eucalyptus globulus (Labill.) Activates Complement Receptor-Mediated Phagocytosis and Stimulates Podosome Formation in Human Monocyte-Derived Macrophages"

_molecules, 2022, doi:10.3390/molecules27113488_

Round 1

Reviewer 1 Report

The authors studied some mechanisms that may explain the immunostimulatory capacity exerted by the essential oil from Eucalyptus globulus (EO) on macrophages. They investigated if the EO-stimulated phagocytosis in macrophages can be due to its major compound eucalyptol.  They also evaluated the effect of EO (i) on microorganism internalization and elimination; (ii) on efferocytosis; (iii) on podosome formation and studied the involvement of complement receptor in EO-stimulated phagocytosis.

The manuscript is clear and carefully organized. The experimental design is well structured and the attained results strongly support the conclusions proposed.

Introduction

Line 73

In addition to eucalyptol, other minor compounds found in essential oil from Eucalyptus globulus must be described.

Discussion

Line 328

Do you have any idea which compounds may be responsible for this protective effect? Can you point to some studies that might support this hypothesis?

Materials and methods

The effect of eucalyptol was only evaluated for phagocytic ability and viability of MDMs. Why this compound was not tested in the other experiments?

Author Response

REFEREE #1 GENERAL COMMENT: The authors studied some mechanisms that may explain the immunostimulatory capacity exerted by the essential oil from Eucalyptus globulus (EO) on macrophages. They investigated if the EO-stimulated phagocytosis in macrophages can be due to its major compound eucalyptol.  They also evaluated the effect of EO (i) on microorganism internalization and elimination; (ii) on efferocytosis; (iii) on podosome formation and studied the involvement of complement receptor in EO-stimulated phagocytosis.

The manuscript is clear and carefully organized. The experimental design is well structured and the attained results strongly support the conclusions proposed.

REFEREE #1 COMMENT: Introduction, Line 73. In addition to eucalyptol, other minor compounds found in essential oil from Eucalyptus globulus must be described.

ANSWER: As recently reported (Moreira et al., 2022; ref #18 of the revised version), other components of EO include some terpenes such α-pinene and limonene, present in percentages around 9% and 2%, respectively, plus more than 40 other minor compounds present in concentrations lower than 0.2% or in traces. We specified this in the revised manuscript and added a reference focused on EO composition (Lanes 76-79; ref #18)

REFEREE #1 COMMENT: Discussion, Line 328. Do you have any idea which compounds may be responsible for this protective effect? Can you point to some studies that might support this hypothesis?

ANSWER: Unfortunately, we do not yet explore this point, and did not find any studies aimed to clarify this issue. We can only hypothesize that some minor components such as α-Pinene and limonene, having antioxidant ability could contribute to this protective effect but, since this is only a speculative consideration, not based on experimental evidence, we did not discuss this point in the manuscript.  

REFEREE #1 COMMENT: Materials and methods. The effect of eucalyptol was only evaluated for phagocytic ability and viability of MDMs. Why this compound was not tested in the other experiments?

ANSWER: As we mentioned in the introduction, the main aim of this work has been to better characterize the immunostimulatory ability exerted by EO on macrophages, for providing further insight into our in vitro and in vivo results previously published (Ref.# 23 of the revised version: Serafino et al., BMC immunology 2008). Taking in mind this aim, the comparative analysis Eucalyptus oil vs eucalyptol simply aimed to confirm that the effects of the whole extract on phagocytic ability and on cytokine/chemokine release by human macrophages could be ascribable to its main constituent, as recently demonstrated for other cells of the innate immunity. We agree with the referee that could be interesting to deepen some questions related to the effects of eucalyptol on the other activities exhibited by EO, and the results from this study are stimulating in this sense. This issue should be eventually the object of future studies aimed to deepen the properties of eucalyptol.

Reviewer 2 Report

In this manuscript Zonfrillo et al. have studied the pharmacological action exerted by the natural product eucalyptus essential oil (EO) in the function of human monocyte-derived macrophages (MDMs). The authors have found that EO and its main active agent eucalyptol, both increase the phagocytic activity of human MDMs without major changes in the release of pro-inflammatory cytokines and chemokines (except for MCP1, whose production was reduced in MDMs by EO treatment). The authors assign to the activity of EO the increased capability of MDMs to clearance phagocytic particles and tumour-derived apoptotic bodies, showing the interesting finding that EO mainly increases macrophage phagocytosis dependent on complement opsonization. In addition, Zonfrillo et al. have shown that EO fosters podosome formation, facilitating the interaction between actin and vinculin and also the expression of the b2 integrin Mac-1.

Major comments

Focussing on the study of human macrophages obtained from blood monocytes, the work by Zonfrillo et al. sheds new light on the reported beneficial actions that EO can play in the treatment of infectious diseases, exploring the mechanisms by which EO exerts some of its functions in the immune response. The main weakness of the current study by Zonfrillo et al. is that some of the results here shown have been yet reported by the authors (Serafino et al., Stimulatory effect of Eucalyptus essential oil on innate cell-mediated immune response, BMC Immunol 2008 Apr 18;9:17. doi: 10.1186/1471-2172-9-17), particularly those results corresponding to the positive effect that EO exerts on the phagocytic activity of macrophages and its negligible activity on the production of pro-inflammatory cytokines compared with LPS treatment. In addition, the current study is largely devoid of functional approaches, which makes difficult to assess the relevance of the new set of results shown.

Some specific points

1.- In general, the in vitro assays have been well performed. However, it seems that monocyte-to-macrophage differentiation has been carried out spontaneously with no addition of any growth factor (e.g. M-CSF) to monocyte cultures. This opens the question of which is the nature of the monocyte-derived macrophages that have been analysed. Are they homogeneous or heterogeneous? The cells that the authors have studied require thus to be characterized in detail. Specific markers of human macrophages subsets should be analyzed by flow cytometry.

2.- Figure 1D: At the concentrations tested, the cell toxicity of eucalyptol is unacceptably high. To confirm that eucalyptol was indeed the main EO-derived active agent that increased phagocytosis of macrophages, cells should be treated with eucalyptol-free EO as negative control.

3.- The study requires additional functional assays. How does EO affect macrophage migration and chemotaxis on integrin ligands? The assessment of this issue may considerable improve the interest and relevance of the manuscript.

4.- Figure 7: Since enhanced integrin levels do not always correlate with increased cell adhesion, the action exerted by EO on integrin function should be also analyzed with the use of activation-specific b2/aM antibodies.

Minor point

1.- Typographic error on the legend to Figure 1 and results on line 136:  The cell viability assay should be labelled as “Figure 1D”.    

Author Response

REFEREE #2 GENERAL COMMENT: In this manuscript Zonfrillo et al. have studied the pharmacological action exerted by the natural product eucalyptus essential oil (EO) in the function of human monocyte-derived macrophages (MDMs). The authors have found that EO and its main active agent eucalyptol, both increase the phagocytic activity of human MDMs without major changes in the release of pro-inflammatory cytokines and chemokines (except for MCP1, whose production was reduced in MDMs by EO treatment). The authors assign to the activity of EO the increased capability of MDMs to clearance phagocytic particles and tumor-derived apoptotic bodies, showing the interesting finding that EO mainly increases macrophage phagocytosis dependent on complement opsonization. In addition, Zonfrillo et al. have shown that EO fosters podosome formation, facilitating the interaction between actin and vinculin and also the expression of the b2 integrin Mac-1.

Major comments

REFEREE #2 COMMENT: Focussing on the study of human macrophages obtained from blood monocytes, the work by Zonfrillo et al. sheds new light on the reported beneficial actions that EO can play in the treatment of infectious diseases, exploring the mechanisms by which EO exerts some of its functions in the immune response. The main weakness of the current study by Zonfrillo et al. is that some of the results here shown have been yet reported by the authors (Serafino et al., Stimulatory effect of Eucalyptus essential oil on innate cell-mediated immune response, BMC Immunol 2008 Apr 18;9:17. doi: 10.1186/1471-2172-9-17), particularly those results corresponding to the positive effect that EO exerts on the phagocytic activity of macrophages and its negligible activity on the production of pro-inflammatory cytokines compared with LPS treatment. In addition, the current study is largely devoid of functional approaches, which makes difficult to assess the relevance of the new set of results shown.

ANSWER: We thank the referee for acknowledging the novelty of our work overall. As it concerns the weakness evidenced by the referee, and particularly referring to “those results corresponding to the positive effect that EO exerts on the phagocytic activity of macrophages and its negligible activity on the production of pro-inflammatory cytokines compared with LPS treatment”, we emphasize that in the experiments performed in the present work, which results are reported in Fig. 1 and 2,  EO has been used not for assessing its activity on phagocytic ability and on cytokines/chemokines released by human macrophages (already demonstrated in our previously published paper Serafino et al., BMC Immunol 2008, ref # 23 of the revised version)  but for the comparative analysis EO vs eucalyptol, aiming to confirm that the effects of the whole extract could be ascribable to its main constituent eucalyptol. Thus, in those experiments, EO was used not as an experimental test but as a “comparative control” (we specified this in the result section, pg. 4, lane 135), while LPS was used as a positive control of macrophage activation, and the new result concerns the effect of eucalyptol.

REFEREE #2 COMMENT on some specific points

REFEREE #2 COMMENT: point 1. In general, the in vitro assays have been well performed. However, it seems that monocyte-to-macrophage differentiation has been carried out spontaneously with no addition of any growth factor (e.g. M-CSF) to monocyte cultures. This opens the question of which is the nature of the monocyte-derived macrophages that have been analysed. Are they homogeneous or heterogeneous? The cells that the authors have studied require thus to be characterized in detail. Specific markers of human macrophages subsets should be analyzed by flow cytometry.

ANSWER: We thank the referee for the comment, but the methods for obtaining the MDMs used in this work has been set up in our lab many years ago [1](Serafino et al, 2014; ref. # 32, online suppl. fig. S1), and the characterization of the differentiated macrophages has been already published as supplementary material in our previous paper (ref. # 32). Specifically, purity and maturation of macrophage cultures were tested by flow cytometry, analyzing physical (forward and side scattering) and immunological parameters (CD14, CD45 and CD44 expression). From this characterization, MDM cultures resulted almost exclusively constituted by the CD45+ and CD14+ monocytic fraction and expressed high levels of CD44 on the cell surface, which are all features of mature macrophages. In any case, the comment of the referee is appropriate, and for addressing his/her suggestion we added in the MM section this detail and the reference of our previous paper (Lanes 402-405)

REFEREE #2 COMMENT: point 2.  Figure 1D: At the concentrations tested, the cell toxicity of eucalyptol is unacceptably high. To confirm that eucalyptol was indeed the main EO-derived active agent that increased phagocytosis of macrophages, cells should be treated with eucalyptol-free EO as negative control.

ANSWER: Really, the cell toxicity of the doses of eucalyptol used (0.0048%, corresponding to 480μM, and 0.0064%, corresponding to 640μM) are not so unacceptably high, since it has been recently reported that, for platelets, eucalyptol was found to be non-toxic to up to 50 µM concentration, but at higher concentrations, like those that we used in our study, it was found to cause cytotoxicity (Alatawi et al, 2021; ref #30) We added this comment and reference in the revised version (Lanes 334-336; ref #30). The molarity of the doses used for eucalyptol has been also added in the MM section of the revised manuscript (Lane 416). Furthermore, regarding the referee's suggestion to use a negative control treatment with eucalyptol-free EO, this, as far as I know, is very difficult to accomplish as eucalyptol makes up to 90% of the essential oil, and I did not find similar works reported in the literature in which this kind of negative control has been used.

REFEREE #2 COMMENT: point 3.  The study requires additional functional assays. How does EO affect macrophage migration and chemotaxis on integrin ligands? The assessment of this issue may considerable improve the interest and relevance of the manuscript.

REFEREE #2 COMMENT: point 4.  Figure 7: Since enhanced integrin levels do not always correlate with increased cell adhesion, the action exerted by EO on integrin function should be also analyzed with the use of activation-specific b2/aM antibodies

ANSWERs to points 3 and 4: We thank the referee for the suggestion, but, as specified in the introduction, the purpose of the present work has been focused on better characterizing the previously demonstrated immunostimulatory ability exerted by EO on macrophages [Serafino et. al, 2008], by investigating different aspects of EO immune-stimulating activity, ranging from the increment of microorganism digestion, the effects on efferocytosis, the receptors involved in the improvement of phagocytic ability, and the effect on podosome formation. Thus, the main aim of this study has been to provide scientific evidence supporting an additional property of this plant extract besides the known antiseptic and anti-inflammatory ones, by exploring a set of effects exerted by EO on different functions related to macrophage-mediated immune response. In the revised version of the manuscript, we better clarify this point in the introduction (pg. 2, lanes 98-100). We agree with the referee that functional assays for exploring the modality by which EO affects macrophage migration and chemotaxis on integrin ligands as well as the action exerted by EO on integrin function are issues worthy of deepening and will be specifically explored in future devoted studies.

REFEREE #2 COMMENT: Minor point

1.- Typographic error on the legend to Figure 1 and results on line 136:  The cell viability assay should be labeled as “Figure 1D”.   

ANSWER: the text has been modified as indicated by the referee.

Round 2

Reviewer 2 Report

Zonfrillo et al. have satisfactorily discussed some of the previous comments and suggestions on the manuscript. However, in view of similar authors' previous results on this topic, that EO is 90% eucalyptol and that the majority of experiments have been performed using EO, important issues still remain to be addressed in order to increase the novelty and interest of this study, particularly those issues concerning experimental work to extend their findings toward functional insights. In this regard, cell adhesion, migration or transmigration experiments are required to better understand the importance of EO in promoting formation of podosomes. Thus this reviewer still considers that any relevant information on this point should considerably increase the significance of the current manuscript.

Author Response

We thank the referee for appreciating our previous answers. To further address the referee’s suggestions, we provide functional evidence demonstrating that EO stimulates macrophage motility and chemotaxis. Specifically, we performed the transwell migration assay to assess whether EO affects the migratory ability of macrophages in response to a chemotactic stimulus, in the absence or in presence of 0.008% EO, and added these results (paragraph 2.6 and Figure 8), as well as the description of the method used (paragraph 4.8), in the revised version of the manuscript.

This manuscript is a resubmission of an earlier submission. The following is a list of the peer review reports and author responses from that submission.

Round 1

Reviewer 1 Report

The manuscript by Zonfrillo et al. presents results from a study aimed at characterizing the effects of eucalyptus oil (EO) and its major constituent, eucalyptol (1,8-cineole), on monocyte-derived macrophages (MDM) in vitro, in order to better understand the immune-stimulatory ability of EO. The authors show that treatment of MDM with EO or equivalent concentrations of eucalyptol stimulates phagocytosis, improves digestion/clearance of internalized material, and stimulates formation of podosomes. They conclude that “EO extract is a potent activator of innate cell-mediated immunity, providing further evidence that supports its use as adjuvant in the treatments of immunosuppressive and infectious diseases, and in tumor chemotherapy.”

Overall, the manuscript is mostly well written and the data are clearly presented. The study represents an extension of this group’s earlier publication (ref. #18), which showed stimulatory effects of EO on MDM phagocytic activity in vitro and in vivo, and on cytokine release. Compared to this earlier study, the current study provides a relatively small increment in further insight.  

The results from the authors’ comparison of the effects of EO side by side to the effects of pure eucalyptol are difficult to interpret, due to the fairly strong cytotoxicity of pure eucalyptol. Furthermore, the extensive toxicity (20%) caused by eucalyptol seems unexpected, as it suggests other constituents in EO suppress its toxic impact when contained in the complex oil. While the authors do refer to this antagonism, they make no effort to reconcile its implication or attempt to investigate it further. For instance, it would be interesting to determine whether toxicity of eucalyptol could be separated from its stimulatory effect on phagocytosis (time course) or the underlying mechanism of toxicity or the reason why it is less toxic as part of the complex oil. Nonetheless, the other data in this manuscript that rely on EO, rather than eucalyptol, can stand on their own to make for a reasonable study.

In several of the figure legends, it is not entirely clear which comparisons were made to calculate p-values. For example, in Figure 2B, the hash/number signs (#) are not explained, which is the more puzzling because they connect a bar with 2 asterisks (**) to one with 3 asterisks (***); also: what is the difference between # and ## and ###?

The authors’ proposition (in the Discussion section) that the effects of EO on podosome formation might suggest that EO could be used for disorders of impaired podosome formation (i.e., WAS and XLT) is too far-reaching and speculative, and should be rephrased. But it would be incredibly interesting if the authors could use their setup to study the impact of EO on podosome formation in MDM from such patients. Similarly, the main conclusion (last half sentence of the Abstract) is more speculation than based on results.

Figure 7 (Western blot) shows increased levels of two integrins after EO treatment. The increase is small and there is no additional data (inclusion of antisense, blocking antibodies) to establish biological relevance of these minor changes.

Please correct minor issues. For example:

Line 14: use “their” instead of “its” (repeated on Line 79). In the figure legends, the authors need to use a comma instead of a period for 1.8-cineole (1,8-cineole). Line 395: use XLT instead of XTL. [h]“eterodimeric”. Etc.

Generally speaking, the term “for the first time” is not necessary and has the potential to be incorrect (who knows, maybe others somewhere in the world have already made the same observation?). It should be avoided, as it adds nothing to the value of this study.

Author Response

 REFEREE #1 COMMENT: The results from the authors’ comparison of the effects of EO side by side to the effects of pure eucalyptol are difficult to interpret, due to the fairly strong cytotoxicity of pure eucalyptol. Furthermore, the extensive toxicity (20%) caused by eucalyptol seems unexpected, as it suggests other constituents in EO suppress its toxic impact when contained in the complex oil. While the authors do refer to this antagonism, they make no effort to reconcile its implication or attempt to investigate it further. For instance, it would be interesting to determine whether toxicity of eucalyptol could be separated from its stimulatory effect on phagocytosis (time course) or the underlying mechanism of toxicity or the reason why it is less toxic as part of the complex oil. Nonetheless, the other data in this manuscript that rely on EO, rather than eucalyptol, can stand on their own to make for a reasonable study. ANSWER: We thank the referee for the comment and suggestions. As we mentioned in the introduction, the main aim of this work has been to better characterize the immunostimulatory ability exerted by EO on macrophages, for providing further insight to our in vitro and in vivo results previously published (Ref.# 22 of the revised version: Serafino et al., BMC immunology 2008, 9, 17, doi:10.1186/1471-2172-9-17). Taking in mind this aim, the comparative analysis Eucalyptus oil vs eucalyptol simply aimed to confirm that the effects of the whole extract on phagocytic ability and on cytokines/chemokines release by human macrophages could be ascribable to its main constituent, as recently demonstrated for other cells of the innate immunity. We agree with the referee that could be interesting to deepen some questions related to the higher toxicity of eucalyptol vs EO, and this will be the object of future studies, but it was not the aim of the present one.

REFEREE #1 COMMENT: In several of the figure legends, it is not entirely clear which comparisons were made to calculate p-values. For example, in Figure 2B, the hash/number signs (#) are not explained, which is the more puzzling because they connect a bar with 2 asterisks (**) to one with 3 asterisks (***); also: what is the difference between # and ## and ###? ANSWER: as we specified in the legend of the figures, the asterisks represent the p values vs the untreated control, while the hash/number signs (#) those vs the p value Eucalyptol vs EO or EO vs LPS etc… In any case, in the revised version of the manuscript, also to reinforce the significance of the results obtained (see below the answer to referee’s comment to Fig 7), we performed the One-way ANOVA + Tukey multiple comparison test or + Dunnett's Comparison Test as appropriate, and modified the significance in the bar graphs of Figs. 1, 2, 3, 4, 6, and 7, also explaining and better clarifying the meanings of both the symbols used (asterisks and #). We thank the referee for highlighting this criticism and allow us to improve the significance of the results and the clarity of the figure legends.

REFEREE #1 COMMENT: The authors’ proposition (in the Discussion section) that the effects of EO on podosome formation might suggest that EO could be used for disorders of impaired podosome formation (i.e., WAS and XLT) is too far-reaching and speculative, and should be rephrased. But it would be incredibly interesting if the authors could use their setup to study the impact of EO on podosome formation in MDM from such patients. Similarly, the main conclusion (last half sentence of the Abstract) is more speculation than based on results. ANSWER: Actually, we have preliminary data obtained on macrophage from two volunteer WAS patients that gave us very interesting results, but we cannot include these results in the manuscript since they will be part of a study that are in ongoing. Perhaps these preliminary results prompted us to speculate on this point. To address the referee’s comments, we rephrased and attenuated the speculative parts discussion. As it concerns the last sentence of the abstract, we did not substantially modify it, since the conclusion is also based on the previously published data (REF# 22, Serafino et al., BMC immunology 2008), for which, as specified, the results reported in the present manuscript are supportive.

REFEREE #1 COMMENT: Figure 7 (Western blot) shows increased levels of two integrins after EO treatment. The increase is small and there is no additional data (inclusion of antisense, blocking antibodies) to establish biological relevance of these minor changes. ANSWER: Since αM and β2 integrins are two of the main structural components of the podosome ring, and together forms the eterodimeric complex CD11b/CD18 (or CR3), the WB analyses of the expression of these two integrins, reported in Fig. 7, have been basically performed to support the results, reported in Figs. 5 and 6, showing that EO stimulates podosome formation and the CR-mediated phagocytosis. Even if the increase in the expression of these two integrins is not dramatically high (up to about two-fold of increment vs CTR), the results were obtained from three independent experiments performed on MDMs from three different donors (we added this detail in the MM and in the legend to figure 7) and the difference EO vs Ctr were nevertheless significant (p<0.05 and higher). In any case, in the revised version of the manuscript, to reinforce and confirm the significance of the results obtained by two-tailed Student’s t test, we performed the One-way ANOVA + Dunnett's Comparison Test, and modified the p values in the bar graphs of Fig. 7 accordingly. Further insight on the biological relevance of these modifications in αM and β2 integrins stimulated by EO, eventually obtained using the methods suggested by the referee, could be object of future studies.  

REFEREE #1 COMMENT: Please correct minor issues. For example:

Line 14: use “their” instead of “its” (repeated on Line 79). In the figure legends, the authors need to use a comma instead of a period for 1.8-cineole (1,8-cineole). Line 395: use XLT instead of XTL. [h]“eterodimeric”. Etc. ANSWER: the text has been modified as suggested by the referee

REFEREE #1 COMMENT: Generally speaking, the term “for the first time” is not necessary and has the potential to be incorrect (who knows, maybe others somewhere in the world have already made the same observation?). It should be avoided, as it adds nothing to the value of this study. ANSWER: the term “for the first time” has been removed

Reviewer 2 Report

The manuscript is about Eucalyptus globulus essential oil's immunostimulant activity, focusing on the mechanism of action, using several in vitro models. The results are very interesting and corroborate the already proved activity of this plant derivative and its traditional use.

Although the manuscript, in general, is well written, the Material Methods(M&M) section should be improved. The first aspect refers to Ethics issues due to the use of blood samples – even though obtained from a transfusional center, human tissues require ethics committee approval. Therefore such information should be included in the M&M section.

Also, considering essential oil can present variable composition depending on the plant geographical origin, used extraction technique, etc., authors should include the details about the essential oil composition. if the authors did the extraction, information about the plant (origin, voucher number, etc.) should be included.

The complete botanical binomial, including the authority, should be included in the title, in the  abstract and in the first time  cited in the main text.

Regarding the assays, it will be interesting inf results from eucalyptol alone is included.

Author Response

The manuscript is about Eucalyptus globulus essential oil's immunostimulant activity, focusing on the mechanism of action, using several in vitro models. The results are very interesting and corroborate the already proved activity of this plant derivative and its traditional use.

REFEREE #2 COMMENT: Although the manuscript, in general, is well written, the Material Methods(M&M) section should be improved. The first aspect refers to Ethics issues due to the use of blood samples – even though obtained from a transfusional center, human tissues require ethics committee approval. Therefore such information should be included in the M&M section. ANSWER: As we wrote in the Informed Consent Statement, no ethical approval or consent to participate was required for this study since it is not a clinical study and the experiments have been performed on PBMCs obtained for anonymous buffy coats. But we erroneously forgot to mention that the PBMCs used for this study had been stored, soon after their isolation from the buffy coats, in our liquid nitrogen cell bank and resuscitated and processed to isolate adhering MDMs, as described in MM, when the experiments were performed (we added this information in the MM section). In the period of the buffy coats providing, no ethics committee approval was required for buffy coats provided anonymously, but at the Blood Transfusion Center of the Policlinico “Tor Vergata”, as a rule, all donors gave their informed consent according to national and international guidelines. We added this information in the MM and in the Informed Consent Statement section of the revised manuscript

REFEREE #2 COMMENT: Also, considering essential oil can present variable composition depending on the plant geographical origin, used extraction technique, etc., authors should include the details about the essential oil composition. if the authors did the extraction, information about the plant (origin, voucher number, etc.) should be included. ANSWER: We did not make the extraction but the essential oil from Eucalyptus globulus used in this study was commercial and purchased from Sigma-Aldrich (St. Louis, Mo, USA; product number W246603, CAS number 84625-32-1). The percent of 1,8-cineole was ≥ 70%, as reported in the product specification sheet. We added this details in the MM section of the revised version.

REFEREE #2 COMMENT: The complete botanical binomial, including the authority, should be included in the title, in the abstract and in the first time cited in the main text. ANSWER: In the firstly submitted version of the manuscript, botanical binomial (Eucalyptus globulus) had been already reported in the title, abstract and main text at the first time cited. Even if this essential oil is widely and from long time studied and the Authority is not usually mentioned in the most of the published papers, to address the referee suggestion we also added the Authority (Labillardière; Labill.) in the abstract and in the main text at the first citation.

REFEREE #2 COMMENT: Regarding the assays, it will be interesting inf results from eucalyptol alone is included. ANSWER: We thanks the referee for the interest and agree with him/her on the fact that could be interesting to deepen and extend all the assays performed with EO to eucalyptol alone, and this will be the object of future studies, but it was not the aim of the present one, that had main goal to better characterize the immunostimulatory ability exerted by EO on macrophages, for providing further insight to our in vitro and in vivo results previously published (Ref.# 22: Serafino et al., BMC immunology 2008, 9, 17, doi:10.1186/1471-2172-9-17). Taking in mind this aim, the comparative analysis Eucalyptus oil vs eucalyptol simply aimed to confirm that the effects of the whole extract on phagocytic ability and on cytokines/chemokines release by human macrophages could be ascribable to its main constituent, as recently demonstrated for other cells of innate immunity.

Reviewer 3 Report

Manuscript ID pharmaceuticals-1544957 Type Article Title Essential oil from Eucalyptus globulus activates complement receptor-mediated phagocytosis and stimulates podosome formation in human monocyte-derived macrophages Authors Manuela Zonfrillo , Federica Andreola , Krasnowska Krystyna Ewa , Gianluca Sferrazza , Pasquale Pierimarchi , Annalucia Serafino *   The manuscript by Zonfrillo et al. describes a detailed study of the effects Eucalyptus essential oil and its main component, eucalyptol, have on macrophages in vitro. The outcomes described in the manuscript are both interesting and important. However, the manuscript is written as much as an advertisement as it is a scientific report. Besides the need to improve the written English and grammar, words like "confirmed" are used in such ways as indicating bias and misunderstanding of Science. The results section contains quite a bit of discussion and conjecture, which inherently leads to redundancy in several parts of the actual discussion section where it belongs. I am recommending this work be reconsidered after major revision since it could use major editing prior to publication.

Author Response

REFEREE #3 COMMENT: The manuscript by Zonfrillo et al. describes a detailed study of the effects Eucalyptus essential oil and its main component, eucalyptol, have on macrophages in vitro. The outcomes described in the manuscript are both interesting and important. However, the manuscript is written as much as an advertisement as it is a scientific report. Besides the need to improve the written English and grammar, words like "confirmed" are used in such ways as indicating bias and misunderstanding of Science. The results section contains quite a bit of discussion and conjecture, which inherently leads to redundancy in several parts of the actual discussion section where it belongs. I am recommending this work be reconsidered after major revision since it could use major editing prior to publication. ANSWER: We thank the referee that appreciated the results described in our work, judging the “outcomes described in the manuscript both interesting and important”. However, it is not clear what he/she means when the referee defines the manuscript “written as an advertisement” or when he/she assert the “words like "confirmed" are used in such ways as indicating bias and misunderstanding of Science”… Besides to do these criticisms, the referee however do not give any (or give very few) practical suggestion on how to improve the manuscript addressing his/her comments. Anyway, attempting to satisfy some of the referee's criticisms, we removed from the results section the redundant parts and shift some of them in the discussion section. The English has been revised with the help of a native speaker corrector and the word “confirmed” has been replaced when appropriate. We hope that the wide revisions we made in the manuscript based on the comments of the other three referees could also satisfy the criticisms raised by this referee.

Reviewer 4 Report

This work is comprehensive but before publication the manuscript will require further revisions, including a proof read for language. The discussion sections are further expanded to include more interpretation of results. Finally, please include statistical analysis section in methods to describe various statistical techniques employed
-How the authors decided to use these concentrations: 0.0048% , 0.0064% , 0.008%. 
- Lines 203 and 296 and more places , please write in passive voice , remove ( we , I,..... ect)
 _ It is required to add more references ,  using the total 28 references is not accepted 

Author Response

REFEREE #4 COMMENT: This work is comprehensive but before publication the manuscript will require further revisions, including a proof read for language. The discussion sections are further expanded to include more interpretation of results. ANSWER: The language has been revised with the help of a native speaker corrector. The Discussion has been revised focusing on discussing the results presented and removing duplication of result description

REFEREE #4 COMMENT: Finally, please include statistical analysis section in methods to describe various statistical techniques employed. ANSWER: In the previously submitted version the statistical analysis section has been already included in the MM (paragraph 4.8), but in the revised version of the manuscript, also to reinforce the significance of the results obtained, we performed, in addition to the two-tailed Student’s t test, the One-way ANOVA + Tukey multiple comparison test or + Dunnett's Comparison Test as appropriate, and include these tests in the paragraph 4.8

REFEREE #4 COMMENT: How the authors decided to use these concentrations: 0.0048%, 0.0064%, 0.008%. ANSWER: As reported in MM section, the concentration of 0.008% EO has been selected in preliminary dose-response experiments previously published [Ref. #22]. In order to obtain comparable doses of treatment between the whole extract and eucalyptol, the latter was used at concentrations equivalent to the 60% and 80% (equal to 0.0048% and 0.0064% of eucalyptol, respectively) of the content of the dose used for the whole extract (0.008%). These percentages of eucalyptol have been chosen assuming that 60% (0.0048%) and 80% (0.0064%) were equivalent to a lower and a higher dose, respectively, than the content of 1,8-cineole reported in the product specification sheet of the commercial EO used in the study (≥ 70%) (see MM). In the revised version of the manuscript, this has been better specified in MM section

REFEREE #4 COMMENT:  Lines 203 and 296 and more places, please write in passive voice, remove (we , I,..... ect). ANSWER: the text has been modified as suggested by the referee.

REFEREE #4 COMMENT: It is required to add more references, using the total 28 references is not accepted. ANSWER: We are a little bit surprised by this comment of the referee, since the Journal do not specify in the Instructions for the Authors a minimum number of references to be cited, also taking in mind that this work is not a review but an original article, and considering that there are various original articles published on the same Journal that have less references cited than in our manuscript. In any case, to address the referee comment, we added in the revised version the one reference concerning a review on phagocytosis of apoptotic bodies or efferocytosis (ref. # 2) and other three references concerning the composition (ref. #11), the antiviral (Ref. # 17) and the immunostimulatory (ref. #20) activities of EO.

References added:

  1. Doran, A.C.; Yurdagul, A., Jr.; Tabas, I. Efferocytosis in health and disease. Nature reviews. Immunology 2020, 20, 254-267, doi:10.1038/s41577-019-0240-6.
  2. Dhakad, A.K.; Pandey, V.V.; Beg, S.; Rawat, J.M.; Singh, A. Biological, medicinal and toxicological significance of Eucalyptus leaf essential oil: a review. J Sci Food Agric 2018, 98, 833-848, doi:10.1002/jsfa.8600.
  3. Mieres-Castro, D.; Ahmar, S.; Shabbir, R.; Mora-Poblete, F. Antiviral Activities of Eucalyptus Essential Oils: Their Effectiveness as Therapeutic Targets against Human Viruses. Pharmaceuticals (Basel) 2021, 14, doi:10.3390/ph14121210.
  4. Sandner, G.; Heckmann, M.; Weghuber, J. Immunomodulatory Activities of Selected Essential Oils. Biomolecules 2020, 10, doi:10.3390/biom10081139.

Round 2

Reviewer 1 Report

It is appreciated that the authors have addressed some of my earlier comments. However, some of the major considerations were not addressed. Therefore, this study remains stuck at a preliminary stage without significant novelty.

The authors’ conclusion that the effects of EO on podosome formation might suggest that EO could be used for disorders of impaired podosome formation (i.e., WAS and XLT) is too far-reaching and speculative. I had suggested that it would be helpful to show initial data obtained from MDM (monocyte-derived macrophages) derived from patients with such disorders, which would make the current study so much stronger. The authors responded that, indeed, they have such preliminary data from two WAS patients, but they “cannot include these results in the manuscript since they will be part of a study that are ongoing.” Well, in that case, it would make sense to hold off on the current manuscript and combine it with the next one, once the ongoing study has been completed. This approach would make more sense than publishing incremental studies. The content of the current manuscript is not sufficiently mature or novel to stand on its own and support the far-reaching claims that are made with regard to benefit of EO for human disorders (podosome disorders, immunosuppressive and infectious diseases, and tumor chemotherapy). There is no relation of the in vitro results shown in this study to any of these conditions. It is not appropriate to provide conclusions based on observations and data that might (or not) be published in the future.  

In my prior critique, I had also stated the following: “Figure 7 (Western blot) shows increased levels of two integrins after EO treatment. The increase is small and there is no additional data (inclusion of antisense, blocking antibodies) to establish biological relevance of these minor changes.” This critique was not addressed; instead, the authors mention that this issue “could be object of future studies.” This therefore remains unresolved. While it is acknowledged that the authors applied statistics to these small differences and calculated a p-value below 0.001, it does nothing to reveal whether or not these statistically significant changes are of any biological relevance. It is quite likely that many other targets (mRNAs, proteins) are affected by EO treatment in a statistically significant fashion, perhaps even to a much larger degree. But with regard to the two integrins, it is not shown that these small (<2-fold) increases exert any impact on podosome function. Based on the authors intent to “deepening some mechanistic aspect” (as declared in the Abstract), a minimal amount of mechanistic studies should be performed. At this stage, the entire study content remains descriptive.

Reviewer 3 Report

Some of the English mistakes from the previous version have been corrected, while others remain and some new ones have been introduced in the revision process. For example, abstract L16 contains a taxonomic authority named in italics that does not need to be, and lacks the family name that would traditionally accompany this information. Another example in the abstract is the modified L31 "additional evidence that support its possible utility as adjuvant" is apparently missing words. Other issues come throughout the manuscript and should be corrected by a professional editor or service.

The now clarified and well-described #/### p-value statistics presented in Figures 1, 3, 4, 5, 6, 7 of the revised manuscript do not seem to be in agreement with the standard deviations shown in error bars in the same locations. Have p-values been evaluated with consideration for only the mean values or the SD from the mean as well? The major variability demonstrated in these results appears to contradict the statistical significance denoted and the "confirmation" of the hypotheses that these authors take from those observations. 

The authors are again reminded about the inappropriate use of the word "confirmed" for when data is observed in support of their hypotheses and obvious biases. This is not acceptable in a scientific report. Advertisements do not belong in scientific journals.

The results section still contains much discussion and conjecture, instead of only results. This does not conform to the policy of the Journal Pharmaceuticals and is not acceptable for publication here. 

I find myself in agreement with other reviewers that this work is inappropriately partitioned from a larger body that the authors have already produced, and its publication in this form would represent a purposeful and unacceptable fragmentation of the literature. This also stands in direct contradiction to the authors' data availability statement (L 600; "All the data produced in this study are reported in this article").

The authors have clearly prepared a significant amount of data on this research topic but, for whatever reason, seem to be opposed to presenting it fully and clearly in the manuscript as submitted and resubmitted. The authors also appear to be set on refusing suggestions by this and other reviewers for improvement of the article. I regretfully suggest that this manuscript be rejected.

Reviewer 4 Report

The manuscript was revised and re-written taking into consideration the reviewer’s suggestions and comments. The current version has been improved. Therefore, It is suitable for publication.